# Use of Low-Dose Tamoxifen to Increase Mammographic Screening Sensitivity in Premenopausal Women

**DOI:** 10.3390/cancers13020302

**Published:** 2021-01-15

**Authors:** Mikael Eriksson, Kamila Czene, Emily F. Conant, Per Hall

**Affiliations:** 1Medical Epidemiology and Biostatistics, Karolinska Institutet, 171 77 Stockholm, Sweden; kamila.czene@ki.se (K.C.); per.hall@ki.se (P.H.); 2Department of Radiology, Perelman School of Medicine, University of Pennsylvania, Philadelphia, PA 19104, USA; emily.conant@pennmedicine.upenn.edu; 3Department of Oncology, Södersjukhuset University Hospital, Karolinska Institutet, 118 83 Stockholm, Sweden

**Keywords:** breast cancer, low dose tamoxifen, mammography screening

## Abstract

**Simple Summary:**

This is the first study to model the effects of tamoxifen on mammographic density and screening sensitivity. Low-dose tamoxifen reduces mammographic density in premenopausal women by, on average, 20%. Potential outcome analyses suggest that a reduction in mammographic density improves mammography screening sensitivity, reduces the proportion of large tumors, and subsequent interval cancers.

**Abstract:**

Increased breast density decreases mammographic sensitivity due to masking of cancers by dense tissue. Tamoxifen exposure reduces mammographic density and, therefore, should improve screening sensitivity. We modelled how low-dose tamoxifen exposure could be used to increase mammographic sensitivity. Mammographic sensitivity was calculated using the KARMA prospective screening cohort. Two models were fitted to estimate screening sensitivity and detected tumor size based on baseline mammographic density. BI-RADS-dependent sensitivity was estimated. The results of the 2.5 mg tamoxifen arm of the KARISMA trial were used to define expected changes in mammographic density after six months exposure and to predict changes in mammographic screening sensitivity and detected tumor size. Rates of interval cancers and detection of invasive tumors were estimated for women with mammographic density relative decreases by 10–50%. In all, 517 cancers in premenopausal women were diagnosed in KARMA: 287 (56%) screen-detected and 230 (44%) interval cancers. Screening sensitivities prior to tamoxifen, were 76%, 69%, 53%, and 46% for BI-RADS density categories A, B, C, and D, respectively. After exposure to tamoxifen, modelled screening sensitivities were estimated to increase by 0% (*p* = 0.35), 2% (*p* < 0.01), 5% (*p* < 0.01), and 5% (*p* < 0.01), respectively. An estimated relative density decrease by ≥20% resulted in an estimated reduction of interval cancers by 24% (*p* < 0.01) and reduction in tumors >20 mm at detection by 4% (*p* < 0.01). Low-dose tamoxifen has the potential to increase mammographic screening sensitivity and thereby reduce the proportion of interval cancers and larger screen-detected cancers.

## 1. Introduction

Breast cancer is the most common cancer in women and mammographic screening has been shown to decrease breast cancer specific mortality [1,2]. However, as the amount of radiographically “white” or “dense” tissue increases, the sensitivity of mammography decreases due to “masking” of cancers by dense breast tissue [3]. A recent FDA issued press release proposed a policy change to improve decision making in mammography screening by providing breast density to referring health care professionals and patients [4]. The use of hormonal replacement therapy is known to increase mammographic density and therefore, lower screening sensitivity [5]. More than half of all premenopausal women attending mammography screening have high breast density corresponding to BI-RADS categories C and D [6]. In addition, women with mammographically dense breasts are more likely to be diagnosed with interval cancers than are women with less dense breasts [7]. Interval cancers are more aggressive than cancers detected at screening [8,9].

A 2.5 mg, low-dose oral preparation of tamoxifen reduces breast density non-inferiorly to the standard therapeutic dose of 20 mg [10] and, is associated with fewer vasomotor side effects compared to the standard dose [10,11]. Current efforts in using standard tamoxifen dosing for breast cancer prevention show a low uptake in the population mainly due to treatment side-effects [12]. It could be that low-dose tamoxifen with fewer side effects may receive more interest in the clinical community and, therefore, increase therapeutic uptake in the population.

To study the effect of low-dose tamoxifen on screening sensitivity, a two-armed randomized trial, placebo vs. tamoxifen, must be performed. The challenge is that approximately 100,000 women would be required to enroll and be followed for at least one screening round. Such a study is for many reasons difficult to pursue.

In this pilot study we model how a mammographic density reduction following intake of low-dose tamoxifen might increase the sensitivity of mammographic screening. In addition, we model how a screening sensitivity change might potentially influence the proportion of interval cancer as well as the proportion of larger tumors at screening diagnosis.

## 2. Materials and Methods

### 2.1. Study Population

The KARMA (Karolinska mammography project for risk prediction of breast cancer) cohort includes 70,877 women recruited at four hospitals in Sweden between 2011 and 2013 [13]. The women were followed until December 2019. The Swedish national screening program invites women aged 40–74 years to screen every second year. At baseline, women responded to a detailed on-line questionnaire on breast cancer risk factors and background characteristics. Participants were linked to the national breast cancer quality register in December 2019 to obtain the date of breast cancer diagnosis, tumor size (±20 mm), and mode of detection. The women were followed for two years after their baseline mammogram. Digital full-field mammograms from mediolateral oblique views of both the right and left breasts were collected consecutively during the study period for the measurement of breast density. The main analysis for this study included the available 28,282 premenopausal women in KARMA. For a sensitivity analysis, the full cohort, also including postmenopausal women, was used.

KARISMA (Karolinska intervention trial) is a double-blind, placebo controlled, randomized six-months non-inferiority dose-determination trial conducted in Sweden between 1 October 2016 and 30 September 2019. Eligible women were randomly assigned in a 1:1 ratio to receive placebo, 1, 2.5, 5, 10, or 20 mg of tamoxifen aiming for 240 women in each arm. The primary endpoint was the change in mammographic density at six months. Digital full-field mammograms were collected at baseline and at six months. A more detailed description is provided in Appendix A.

### 2.2. Mammographic Density Measurement

We measured the percent of mammographically dense tissue using the fully automated STRATUS tool [14]. One of four BI-RADS breast density categories; almost entirely fatty (A), scattered areas of fibro-glandular (B), heterogeneously dense (C), and extremely dense (D) [15] was also estimated using the STRATUS tool. For women without a cancer diagnosis, baseline mammograms were used for density estimates; for women diagnosed with cancer, the most recent screening mammogram prior to the cancer diagnosis was used. Mammographic response to tamoxifen was measured using the KARISMA trial as the relative change in area density at six months compared with density measured at baseline. Mammographic density was measured as the average of left and right breast densities in KARISMA and KARMA.

### 2.3. Statistical Analysis

#### 2.3.1. Potential Mammographic Density Response to Low-Dose Tamoxifen

The distribution of density responses in the 2.5 mg KARISMA arm was used as the reference of density response to low-dose tamoxifen we would expect to observe in the premenopausal KARMA women, if they had been exposed to low-dose tamoxifen. An example of a density response to tamoxifen of a woman participating in the KARISMA trial is presented in Appendix A. The KARISMA study showed that density response to tamoxifen did not depend on background characteristics of the study participants after adjustment for multiple comparison (Appendix A) [10,16]. The KARISMA density responses after exposure to 2.5 mg of tamoxifen could, therefore, be applied to the KARMA women using a random distribution. For example, a tamoxifen induced relative density decrease of 10% seen in a KARISMA participant was applied to a random selected woman in KARMA. The 10% density decrease was subtracted from the measured density of the KARMA woman. Each density response in KARISMA that was applied to a random selection of KARMA women created a cluster of density response women in KARMA. Therefore, all statistical tests in this study were performed using robust regression to take the clustered density responses into account [17].

The density responses in the KARMA premenopausal women defined the potential outcome that we would observe if the women were exposed to 2.5 mg of tamoxifen for six months prior to the time of mammographic imaging [18]. The KARMA women with potential tamoxifen exposure and density responses are referred to as the exposed group. The unexposed group refers to the premenopausal KARMA women and their actual density measures.

A tamoxifen density responder was defined as a woman with a relative density decrease greater than or equal to the median decrease in density across the entire exposed group [10,19]. In the KARISMA trial, the median decrease in density across all premenopausal women was ~20%. Additional density responder decrease cut-offs were explored in the range between 10% to 50% in the study. The density response effect on screening sensitivity and tumor size were estimated by the STRATUS-estimated, BI-RADS density categories A + B, C, and D. BI-RADS A and B were combined due to the few premenopausal women in category A.

#### 2.3.2. Screening Sensitivity

Screening sensitivity was defined as the percentage of screen-detected cancers in the sum of all screen-detected and interval cancers. Sensitivity was estimated after adjustment for year of examination to compare screen-detected and interval cancers occurring in women attending screening the same year. A screen-detected cancer was defined as a cancer detected as a result of a screening visit, that is, within three months of the screen. An interval cancer was defined as a breast cancer not diagnosed as a result of screening, but at least three months after a negative screen, but before the date of the next scheduled screening [20]. The mean time between screen and diagnosis was 1.1 years. All cancers with a mammogram available within two years prior to diagnosis were included.

A logistic model was used to estimate the screening sensitivity dependency on the baseline breast density of the study participants [21]. The model was used to predict screening sensitivity in the exposed group. The probability of identifying breast cancers at increased screening sensitivity per percentage density decrease was estimated based on the model. The percentages of screening sensitivity changes were presented by the degree of density response.

#### 2.3.3. Tumor Size

High mammographic density is associated with large tumor size at diagnosis [22]. A logistic model was fitted to estimate the probability of having a >20 mm invasive tumor in the unexposed group based on the density level at baseline. The model was used to predict the probability of having a >20 mm invasive tumor detected at screening in the exposed group, i.e., after a tamoxifen induced density response.

#### 2.3.4. Interval Cancers

The reduction of interval cancers as a consequence of tamoxifen induced breast density reduction was calculated. The calculation was based on the arithmetic transform in Equation (1):IC = (SC/S) − SC (from the formula S) = SC/(SC + IC)(1)
where S is sensitivity, SC is screening detected cancer, and IC is interval detected cancer. That is, sensitivity equals screen-detected cancers divided by the sum of screen-detected and interval cancers.

The number of interval cancers per 100,000 screened women was estimated after adjustment for year of mammogram and age standardization. The age standardization was performed in five-year groups [23]. The reference population was all women who were invited to participate in the national screening program at the hospitals in the south and middle regions of Sweden, where women in the KARMA cohort were recruited.

#### 2.3.5. Sensitivity Analysis

A sensitivity analysis was performed to estimate the reduction in interval cancers using the density responses in the KARISMA trial from the 2.5, 5, 10, 20 mg arms combined as the reference of density response to tamoxifen.

A second sensitivity analysis was performed using the full KARMA cohort including both postmenopausal women and premenopausal women to investigate if menopausal status affects the dependence of screening sensitivity and tumor size on mammographic density. This was performed because some premenopausal women would be expected to transition to a postmenopausal status during the study period resulting in an expected decrease in mammographic density, thus potentially modifying the screening sensitivity and tumor size associations with mammographic density.

All statistical analyses were performed using SAS 9.4.

## 3. Results

### 3.1. Study Population Baseline Characteristics

A total of 517 cancers in 28,282 women were diagnosed with breast cancer in the premenopausal KARMA cohort; 56% (287/517) of the women had screen-detected and 44% (230/517) had interval cancers (Table 1). Interval cancers were more likely invasive (91%) compared to screen-detected cancers (78%). Percent mammographic density was higher in women with interval cancers (39.1%) compared to women with screen-detected cancers (31.6%). A family history of breast cancer was more common among screen-detected (20%) and interval cancer women (25%) compared to women without a cancer diagnosis (12%).

### 3.2. Mammographic Density Response to Low-Dose Tamoxifen

The premenopausal KARISMA women randomized to 2.5 mg of tamoxifen had 39.9 cm^2^ median baseline dense area. After six months of tamoxifen therapy the mean relative dense area decrease was 15.4% [10]. Initial mammographic density level at baseline did not affect the magnitude of density response to tamoxifen (Appendix A). Applying the density change after exposure to 2.5 mg of tamoxifen to the KARMA women, a mean dense area of 32.4 cm^2^ and a mean relative dense area decrease of 15.4% was estimated for the non-cancer KARMA women (Appendix A). A total of 72% of the exposed women experienced a relative density decrease of ≥10%. The corresponding values for a ≥20%, ≥30%, and ≥50% decrease was 55%, 27%, and 11%, respectively (Appendix A).

### 3.3. Screening Sensitivity

The estimated BI-RADS category-dependent sensitivity of mammography, before and after exposure to 2.5 mg of tamoxifen, is presented in Figure 1. There was no estimated influence of tamoxifen on sensitivity in the BI-RADS A category, the sensitivity was 76% in both the exposed and unexposed groups (Figure 1, Table 2). There was a 5% improvement in sensitivity (51% vs. 46%, *p* < 0.01) seen among women with extremely dense breasts (category D). The lowest estimated sensitivity improvement was found at approximately 60% measured mammographic density (Figure 1). Women with less dense breasts (BI-RADS A + B) had an estimated influence of tamoxifen on sensitivity of 2% increase (*p* < 0.01). For women at high density (BI-RADS C + D) the corresponding estimate was 4% increase in sensitivity (*p* < 0.01).

Figure 2 presents the change in BI-RADS category dependent sensitivity as a function of density decrease. The estimated sensitivity in BI-RADS A + B increased from 71% to 74% (*p* < 0.01) in the group of women with a relative density decrease of ≥20% (Figure 2, Table 3). The corresponding estimated sensitivity increases for BI-RADS categories C and D were 53–60% and 46–53%, respectively. The estimated combined sensitivity after tamoxifen exposure, summed across all women (BI-RADS A–D), increased from 55–62% (Figure 2, Table 3).

### 3.4. Tumor Size

Figure 3 presents the estimated probability of having an invasive cancer >20 mm in relation to the estimated relative density decrease in breast density after exposure to tamoxifen. A trend (*p* < 0.001) of lower probabilities of >20 mm invasive tumors was observed in women with BI-RADS C and D densities (Figure 3, Table 4). Specifically, in this group of women with dense breasts, the probability of having a >20 mm invasive tumor was reduced by 4% (*p* < 0.01) if the reduction in breast density was ≥20% following tamoxifen exposure.

### 3.5. Interval Cancers

Table 5 presents the number of interval cancers per 100,000 age-standardized screened premenopausal women in the KARMA cohort and the estimated change in number of interval cancers as a consequence of tamoxifen exposure. The estimated number of interval cancers was recalculated after adding the tamoxifen induced density responses by ≥10%, ≥20%, ≥30% and ≥50% relative decreases. The number of interval cancers was lowest in the BI-RADS A + B category and highest in the C category. A relative density decrease by ≥50% was estimated as having a greater impact on the reduction of interval cancers than a ≥10% relative decrease. Adding up all interval cancers in the A–D categories (*N* = 813), a ≥20% relative density decrease after exposure to low-dose tamoxifen, reduced the estimated number of interval cancers by 192/10^5^ (24%, *p* < 0.01; Table 5, Appendix A).

### 3.6. Sensitivity Analysis

Similar results were found in the estimated sensitivity analysis assuming a density decrease in the KARMA cohort based on using the KARISMA trial density responses from the 265 women in the 2.5, 5, 10, 20 mg arms as the reference for response to tamoxifen (Appendix A). Menopausal status did not affect the screening sensitivity dependence of mammographic density (*p* = 0.37) nor the tumor size dependence of mammographic density, *p* = 0.79 (Appendix A).

## 4. Discussion

We have previously shown that after exposure to 2.5 mg of tamoxifen for six months, the average relative density decrease was 15.4% in the KARISMA trial, a density change non-inferior to the density change seen after exposure to 20 mg standard dose of tamoxifen. Based on the KARISMA trial results, we estimated that after six months exposure to 2.5 mg of tamoxifen, the sensitivity of screening mammography would increase by 4% for the KARMA women. A ≥20% relative decrease in density was seen in 55% of the population. In this group of women, we estimated that mammographic sensitivity would increase by 7% and the number of interval cancers would decrease by 24%, and the number of large screen-detected breast cancers (size >20 mm) would decrease by 4%. In addition, we estimated that the number of such large, screen-detected breast cancers would decrease nearly by one half if breast density decreased by 50%.

The KARISMA six-month randomized controlled dose-determination trial was used as the reference for density responses to tamoxifen. The density responses should, therefore, be reliable. No association was found in linear and quadratic models between density response to tamoxifen after six months and background risk factors assessed at baseline including family history of breast cancer. We could, therefore, apply the density response distribution from the KARISMA trial to the KARMA women using a random distribution of relative density decreases. The density responses were applied to the baseline mammograms of women without a cancer diagnosis and to the last prior screening mammograms of women diagnosed with breast cancer in the KARMA cohort. The KARMA women in our model were therefore assumed to have had a six-month tamoxifen therapy prior to the mammographic examination based on density responses observed in KARISMA.

It has been shown that hormonal replacement therapy increases mammographic density and reduces screening sensitivity. In this study we modelled the inverse effect of a tamoxifen induced density reduction. We studied the potential effect of low-dose tamoxifen on a decrease in mammographic density and the resultant increase in screening sensitivity. The estimated relative dense area reduction in the exposed group was ~15% (6 cm^2^) after six months of 2.5 mg tamoxifen exposure. In comparison, the natural annual decrease of mammographic dense area due to involution is approximately 1 cm^2^ [24].

Interval cancers are more often found in women with dense breasts and such cancers tend to be larger and more aggressive tumor subtypes [25]. Therefore, women diagnosed with interval cancers have a worse prognosis than women diagnosed with a screen-detected breast cancer [7,9]. Our results suggest that a relative density decrease by ≥20% has the potential to reduce the number of interval breast cancer by 24%. In women with extremely dense breasts (BI-RADS category D), an estimated decrease of interval cancers of up to 42% could potentially be achieved.

In the KARISMA trial, approximately 50% fewer severe vasomotor side effects were reported in the 2.5 mg arm compared with women randomized to the standard, 20 mg dose of tamoxifen [10]. The study showed that no other menopausal-like gynecological, sexual, or musculoskeletal symptoms were dependent on tamoxifen dose. These findings suggest that a low dose of tamoxifen has the potential to increase uptake in the population that is at risk of masking of breast cancer on mammography by dense breast tissue [12].

The biology behind the breast density response to tamoxifen is not well known. Tamoxifen blocks estrogen receptors in the cell nucleus from binding circulating estrogens and from initiating transcription, gene expression, and cell proliferation [26]. As a consequence, tamoxifen reduces the amount of fibro-glandular tissue which is mostly (>90%) composed of stromal tissue [27]. It is therefore plausible that the decrease of radio-dense tissue is mainly reflecting a decrease of stroma [28].

A recent study has shown that higher mammographic density is associated with all forms of breast cancer and not with a specific tumor subtype [8]. We show that a density decrease after tamoxifen exposure is associated with an estimated improvement in identifying smaller tumors (size ≤ 20 mm) at screening. The unmasking effect of breast cancers through decreases in breast density, therefore, has the potential to improve the early detection of several tumor subtypes and therefore, possibly improve patient outcomes. Although tamoxifen is associated with the prevention of estrogen positive cancers specifically, [19] the reduction in fibro-glandular tissue from tamoxifen exposure could improve the early detection of cancers, including estrogen negative receptor status or other tumor characteristics. A tamoxifen induced density decrease could thereby, also improve survival of aggressive breast cancer [9]. Studies have shown that tamoxifen reduces the size of estrogen positive tumors in the neoadjuvant setting [29]. In the preclinical stage of a tumor, the tamoxifen effect on tumor size could lead to that low-grade, non-fatal estrogen positive tumors remain below the detection capability of mammographic screening.

We should note that in our study there is a potential underestimation of the improvement in early detection in our tamoxifen exposed group. In this study, we estimated the percentage of interval cancers that potentially could be identified at their prior screen therefore, improving screening sensitivity. A reduction of mammographic density by tamoxifen could also lead to screen detected cancers being detected earlier on their prior routine screening. In addition, there is a potential overestimation of the number of breast cancers in our tamoxifen exposed group since tamoxifen therapy could also reduce the number of cancers in the exposed group due to the known, preventative effects of the drug on estrogen-positive tumors [19,30].

A recent study showed that a low dose of 5 mg tamoxifen decreased recurrence of breast intraepithelial neoplasia by approximately 50% [11]. It could be that even at low doses, tamoxifen may decrease breast cancer incidence in addition to reducing mammographic density and increasing screening sensitivity. These benefits of low dose tamoxifen, in combination with the preventive effects of decreasing the rate of interval cancers, can only improve the long-term outcomes of breast cancer screening.

Both the weakness and strength of this study are in the study design. We combined the tamoxifen density response effect in the randomized clinical KARISMA trial with screening sensitivity in the KARMA screening cohort. Using this combination, we estimated how the density response to tamoxifen could affect outcomes in a mammographic screening setting. In addition, the women in the KARMA study were not actually administered tamoxifen and we could, therefore, not estimate any preventive effect of low-dose tamoxifen. The women in our study had digital mammograms and we could, therefore, not assess the corresponding screening sensitivity effect based on tomosynthesis mammograms. A future intervention study is needed to estimate the actual reduction of breast cancer incidence among women using low-dose tamoxifen as well as the rate of adherence to the drug exposure for women in organized and opportunistic screening settings.

## 5. Conclusions

In conclusion, we modelled the effect of using low-dose tamoxifen for decreasing breast density and increasing the sensitivity of mammographic screening among premenopausal women. A tamoxifen dose of 2.5 mg over six months has the potential to decrease the number of interval cancers and large screen-detected breast cancers. If our assumptions hold, this strategy has the potential to substantially improve breast cancer screening outcomes as well as patient prognoses. A prospective follow-up study is warranted.

## Figures and Tables

**Figure 1 cancers-13-00302-f001:**
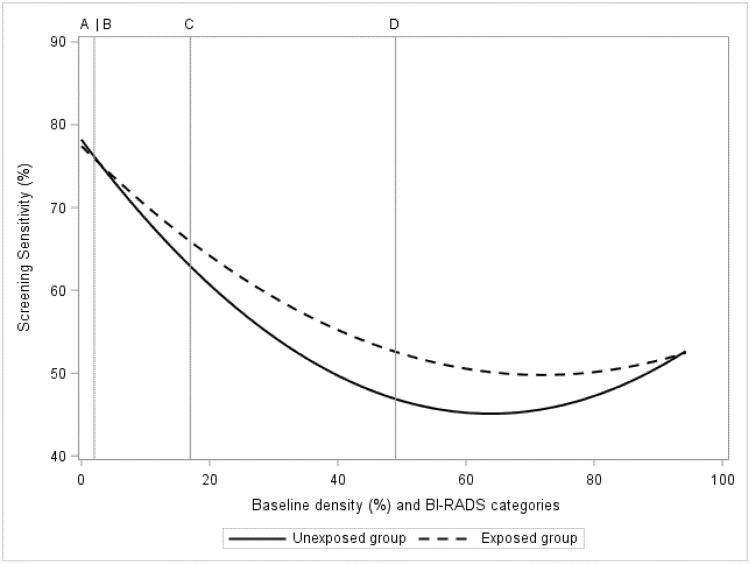
In the KARMA exposed and unexposed groups, the figure presents screening sensitivity by mammographic density at baseline. Baseline mammographic density is presented as a regression plot and as categories of computer-generated BI-RADS density categories A, B, C, D.

**Figure 2 cancers-13-00302-f002:**
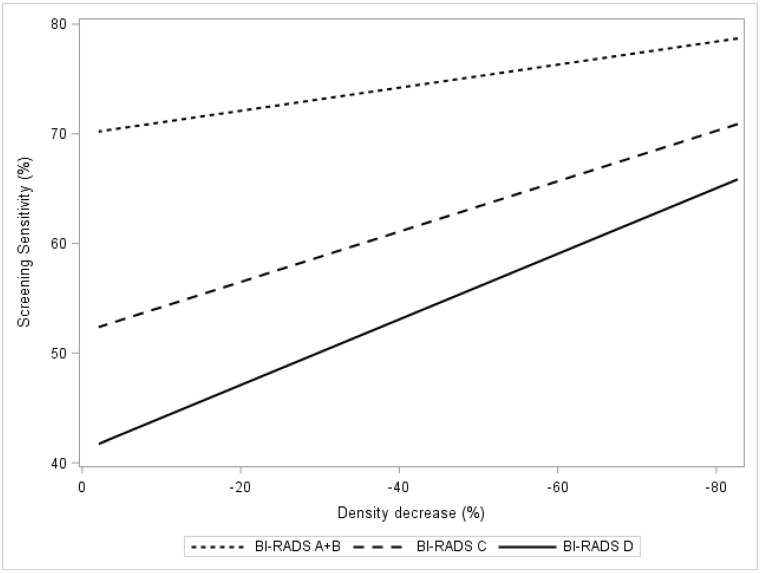
In the KARMA exposed group, the figure presents screening sensitivity by percent mammographic density. Regression plots present breast density response as relative density decrease 0 to 100% stratified by computer-generated BI-RADS density categories A + B, C, D.

**Figure 3 cancers-13-00302-f003:**
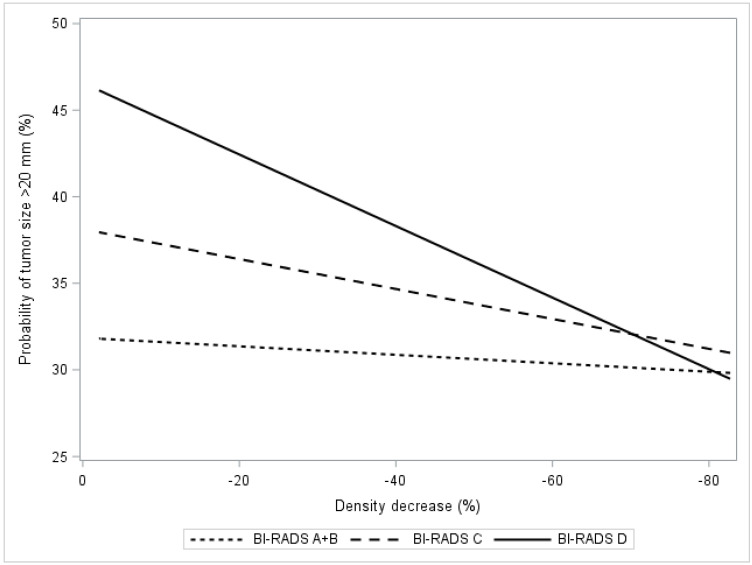
In the KARMA exposed group, the figure presents the probability of being diagnosed with a large tumor (size >20 mm vs. ≤20 mm) in relation to percentage relative density decrease. The relation is presented as regression plots stratified by computer-generated BI-RADS categories A + B, C, D.

**Table 1 cancers-13-00302-t001:** Baseline characteristics of the 28,282 premenopausal women in the KARMA cohort.

Study Participant Characteristics	Non-Breast Cancers	Screen Detected Cancers	Interval Detected Cancers
Number of women	27,765	287	230
Age at baseline, mean (SD)	45.0 (4.0)	46.6 (4.1)	45.1 (4.2)
Invasive breast cancer, %	-	78	91
Tumor of size >20 mm, %	-	36	42
Age at breast cancer diagnosis, mean (SD)	-	49.9 (4.6)	49.1 (4.6)
BMI, mean (SD)	24.9 (4.3)	25.1 (4.0)	24.3 (3.6)
Age at menarche, mean (SD)	13.0 (1.5)	12.9 (1.3)	12.9 (1.6)
Parity, %	87	85	88
Age at first birth, mean (SD)	28.7 (5.2)	29.3 (4.7)	29.6 (4.9)
Current use of hormone replacement therapy, %	5	7	8
Regular smoking during last year, %	10	12	10
Regular alcohol drinking during last year, grams/week	44.6 (54.6)	50.5 (64.6)	44.6 (48.7)
Breast cancer in family 1st degree, %	12	20	25
Percent mammographic density, mean (SD)	31.4 (20.9)	31.6 (21.4)	39.1 (20.3)
Mammographic dense area cm^2^, mean (SD)	38.2 (26.5)	42.5 (30.3)	49.2 (29.0)
Distribution of BI-RADS categories ^1^			
A	6	6	2
B	25	23	17
C	49	51	47
D	21	20	34

SD = standard deviation. ^1^ Computer-generated BI-RADS breast composition categorization. KARMA = Karolinska mammography project for risk prediction of breast cancer. In the KARMA cohort the table presents baseline characteristics of the premenopausal women (*N* = 28,282) stratified by women without breast cancer and breast cancer cases by screening and interval cancer status.

**Table 2 cancers-13-00302-t002:** Screening sensitivity in the unexposed and exposed groups by BI-RADS classification of mammographic breast density at baseline.

Screening Sensitivity, Category Mean (%)	BI-RADS Density Category				Low	High	Low +
	A	B	C	D	A + B	C + D	High
Unexposed group	76	69	53	46	70	51	56
Exposed group	76	71	58	51	72	55	60
Difference	0	2	5	5	2	4	4
*p*-value	0.35	<0.01	<0.01	<0.01	<0.01	<0.01	<0.01

Low density is defined as categories A + B and high density as C + D.

**Table 3 cancers-13-00302-t003:** Screening sensitivity by percentage relative mammographic density decrease in the exposed group in relation to the unexposed group.

Screening Sensitivity, Category Mean (%)	Unexposed Group	Relative Density Decrease (%)			
		≥10	≥20	≥30	≥50
BI-RADS Density Category	-	Exposed Group			
A + B	71	73	74	75	77
C	53	59	60	63	67
D	46	51	53	56	61
A to D combined	55	61	62	63	68
-	-	Difference in percent compared to unexposed group			
A + B	ref.	2	3	4	7
C	ref.	6	7	10	14
D	ref.	5	7	10	15
A to D combined	ref.	6	7	8	13

The table presents mean screening sensitivity tabulated by relative density decrease in range ≥10% to ≥50%. The unexposed group is included as the reference (ref.). Screening sensitivity increased by increasing relative density response in relation to the unexposed group, *p* < 0.01.

**Table 4 cancers-13-00302-t004:** Probability of being diagnosed with a large tumor (size >20 mm) by percentage relative mammographic density decrease in the exposed group in relation to the unexposed group.

Probability of Tumor Size	Unexposed	Relative Density Decrease (%)				*p*-Value
>20 mm, Category Mean (%)	Group	≥10	≥20	≥30	≥50	Trend
BI-RADS A + B	33	32	31	30	30	0.14
BI-RADS C	39	35	35	34	33	<0.01
BI-RADS D	42	39	38	36	33	<0.01

The table presents mean tumor size probability tabulated by relative density response in range ≥10% to ≥50%. The unexposed group was included as the reference.

**Table 5 cancers-13-00302-t005:** Number of interval cancers per 100,000 age standardized screened premenopausal women in the unexposed group and in the exposed group by percentage of relative density decrease.

Number of Interval Cancers	Unexposed Group (*N*)	Relative DensityDecrease, %			
		≥10	≥20	≥30	≥50
BI-RADS Density Category	-	Exposed Group (*N*)			
A + B	155	107	102	100	88
C	382	356	339	299	253
D	276	197	180	160	129
A to D combined	813	660	621	559	470
-	-	Difference Compared to Unexposed Group, *N* (%)			
A + B	ref.	−48 (−31)	−53 (−34)	−55 (−35)	−67 (−43)
C	ref.	−26 (−7)	−43 (−11)	−83 (−22)	−129 (−34)
D	ref.	−79 (−29)	−96 (−35)	−116 (−42)	−147 (−53)
A to D combined	ref.	−153 (−19)	−192 (−24)	−254 (−31)	−343 (−42)

In the KARMA exposed group, the table presents the number of interval cancers per 100,000 age standardized screened premenopausal women together with the change in numbers of interval cancers by percentage relative mammographic density decrease. Relative density decrease was stratified by computer-generated BI-RADS categories A + B, C, D. The unexposed group was included as the reference (ref.). Number of interval cancers was reduced by the relative density response ranging from ≥10% to ≥50% in relation to the unexposed group, *p* < 0.01.

## Data Availability

Data is available upon reasonable request to karmastudy.org. The data are not publicly available due to the fact that each request has to be evaluated in accordance with international and national regulations, in addition to the consent provided by each participant.

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
