# Peer review of "Use of Low-Dose Tamoxifen to Increase Mammographic Screening Sensitivity in Premenopausal Women"

_cancers, 2021, doi:10.3390/cancers13020302_

Round 1
Reviewer 1 Report
This is a very nice, clearly written paper that describes a modeling study of the effects of Tamoxifen on mammographic sensitivity and impact on interval cancer detection. The study does an excellent job of using the results from pervious studies (KARISMA and KARMA) to model the effects of low-dose Tamoxifen use.
I have only 2 minor comments:
1. When I first read the Simple Summary, I got the impression that this was a clinical trial. Perhaps the word "estimate" in the first sentence could be changed to "model".
There were some scattered grammatical errors (dropped verbs, etc). Please proof read the manuscript one more time.
Author Response
We are thankful for the comments to our submission and have made all alterations as suggested.
Reviewer comments: This is a very nice, clearly written paper that describes a modeling study of the effects of Tamoxifen on mammographic sensitivity and impact on interval cancer detection. The study does an excellent job of using the results from pervious studies (KARISMA and KARMA) to model the effects of low-dose Tamoxifen use.
I have only 2 minor comments:
- When I first read the Simple Summary, I got the impression that this was a clinical trial. Perhaps the word "estimate" in the first sentence could be changed to "model".
Response R1-1. Thank you for your very encouraging response. We now use the term “model” already in the Simple summary.
There were some scattered grammatical errors (dropped verbs, etc). Please proof read the manuscript one more time.
Response R1-2. We have now carefully proofread the manuscript and have now removed “dropped verbs”, etc. We updated the introduction, material and methods, results, discussion, and conclusion.
Reviewer 2 Report
The topic is real interesting and a potential use of this drug in population could help cancer detection.
However, the study date is between 2011 and 2013 and no DBT comparison or FFDM/US is considered.
Is tamoxifen use more useful than DBT? I remind authors to include consideration about this side of breast cancer screening.
Author Response
We are thankful for the comments to our submission and have made all alterations and commented as suggested.
Reviewer comments: The topic is real interesting and a potential use of this drug in population could help cancer detection. However, the study date is between 2011 and 2013 and no DBT comparison or FFDM/US is considered. Is tamoxifen use more useful than DBT? I remind authors to include consideration about this side of breast cancer screening.
Response R2-1. We appreciate the intriguing and speculative question. We have now clarified in the Materials and Methods section 2.1 that the women were recruited between 2011 and 2013 but were followed till December 2019. The section now reads “The KARMA (Karolinska mammography project for risk prediction of breast cancer) cohort includes 70,877 women recruited at four hospitals in Sweden between 2011 and 2013. [13] The women were followed till December 2019.”
In addition, we have clarified in the Discussion that the KARMA cohort includes women that had mammograms based on digital mammography, but not tomosynthesis. We updated the last section as follows: “The women in our study had digital mammograms and we could therefore not assess the corresponding screening sensitivity effect on tomosynthesis mammograms”.
For this reason, we were not able to investigate this interesting question further.
Reviewer 3 Report
Dear sirs/madams:
On reviewing the attached manuscript, I find the premise and the conclusions to be compelling evidence of the potential for low dose tamoxifen to increase the sensitivity of screening mammography, and to find smaller cancers at screening, both of which have the potential to improve outcomes for women with breast cancer. I did find the “statistical analysis” section of the paper to be quite dense and confusing to me, perhaps because I am not an expert the area of statistical analysis. If it would be possible for the authors to rewrite this section (lines 99-123) with more clarity, I think it would benefit the overall understanding of the data supporting their conclusions. I did find one typographical error in line 267, which I noted in the attached version of the manuscript.Author Response
We are thankful for the comments to our submission and have made all alterations as suggested.
Reviewer comments: Dear sirs/madams: On reviewing the attached manuscript, I find the premise and the conclusions to be compelling evidence of the potential for low dose tamoxifen to increase the sensitivity of screening mammography, and to find smaller cancers at screening, both of which have the potential to improve outcomes for women with breast cancer.
I did find the “statistical analysis” section of the paper to be quite dense and confusing to me, perhaps because I am not an expert the area of statistical analysis. If it would be possible for the authors to rewrite this section (lines 99-123) with more clarity, I think it would benefit the overall understanding of the data supporting their conclusions.
Response R3-1. Thank you for your positive response and the need to clarify the 2.3.1 section. We fully agree with that and now updated this section as follows:
The distribution of density responses in the 2.5 mg KARISMA arm was used as the reference of density response to low-dose tamoxifen we would expect to observe in the premenopausal KARMA women, if they had been exposed to low-dose tamoxifen. An example of a density response to tamoxifen of a woman participating in the KARISMA trial is presented in Supplementary Figure 1. The KARISMA study showed that density response to tamoxifen did not depend on background characteristics of the study participants after adjustment for multiple comparison (Supplementary Table 1). [10,16] The KARISMA density responses after exposure to 2.5 mg of tamoxifen could therefore be applied to the KARMA women using a random distribution. For example, a tamoxifen induced relative density decrease of 10% seen in a KARISMA participant was applied to a random selected woman in KARMA. The 10% density decrease was subtracted from the measured density of the KARMA woman. Each density response in KARISMA that was applied to a random selection of KARMA women created a cluster of density response women in KARMA. Therefore, all statistical tests in this study were performed using robust regression to take the clustered density responses into account. [17]
The density responses in the KARMA premenopausal women defined the potential outcome that we would observe if the women were exposed to 2.5 mg of tamoxifen for six months prior to the time of mammographic imaging. [18] The KARMA women with potential tamoxifen exposure and density responses are referred to as the exposed group. The unexposed group refers to the premenopausal KARMA women and their actual density measures.
I did find one typographical error in line 267, which I noted in the attached version of the manuscript.
Response R3-2. Thank you very much for the detailed proof reading. We now corrected the typos and the text now reads is as follows:
Interval cancers are more often found in women with dense breasts and such cancers tend to be larger and more aggressive tumor subtypes. [25] Therefore, women diagnosed with interval cancers have a worse prognosis than women diagnosed with a screen-detected breast cancer. [5,8] Our results suggest that a relative density decrease by ≥20% has the potential to reduce the number of interval breast cancer by 24%. In women with extremely dense breasts (BI-RADS category D), an estimated decrease of interval cancers of up to 42% could potentially be achieved.